# Survival Outcomes Associated with the Location of BRCA Mutations in Ovarian Cancer: A Systematic Review and Meta-Analysis

**DOI:** 10.3390/cancers17101661

**Published:** 2025-05-14

**Authors:** Ji Hyun Kim, Hyung Joon Yoon, Hyeong In Ha, Eun Taeg Kim, Dongkyu Eugene Kim, Sangeon Kim, Jae Kyung Bae, Myong Cheol Lim

**Affiliations:** 1Center for Gynecologic Cancer, Research Institute and Hospital, National Cancer Center, Goyang 10408, Republic of Korea; gynlittle@gmail.com (J.H.K.); dkim728@uwo.ca (D.E.K.); skim249@mgh.harvard.edu (S.K.); 14448@ncc.re.kr (J.K.B.); 2Department of Obstetrics and Gynecology, Pusan National University Hospital, Busan 49241, Republic of Korea; obgyoon@pusan.ac.kr; 3Biomedical Research Institute, Pusan National University Hospital, Busan 49241, Republic of Korea; 4Department of Obstetrics and Gynecology, Pusan National University College of Medicine, Busan 49241, Republic of Korea; hi126908111@gmail.com; 5Department of Obstetrics and Gynecology, Pusan National University Yangsan Hospital, Yangsan 50612, Republic of Korea; 6Department of Obstetrics and Gynecology, Kosin University Gospel Hospital, Kosin University College of Medicine, Busan 49267, Republic of Korea; dikei03@naver.com; 7Department of Cancer Control and Policy, National Cancer Center Graduate School of Cancer Science and Policy, National Cancer Center, Goyang 10408, Republic of Korea; 8Rare & Pediatric Cancer Branch and Immuno-Oncology Branch, Division of Rare and Refractory Cancer, Research Institute and Hospital, National Cancer Center, Goyang 10408, Republic of Korea

**Keywords:** BRCA1, BRCA2, exon 11, ovarian cancer, survival outcomes, meta-analysis

## Abstract

The location of BRCA 1/2 mutations may influence clinical outcomes in patients with ovarian cancer; however, a definitive pattern has not yet been identified. Therefore, we performed this systematic review and meta-analysis. We identified that BRCA2 exon 11 mutations demonstrated a PFS advantage, whereas BRCA1 exon 11 mutations had no significant effect on PFS and OS. These findings emphasize the significance of mutation-specific treatment strategies and indicate the need for further prospective studies to optimize therapeutic approaches.

## 1. Introduction

*BRCA1*- and *BRCA2*-associated hereditary breast and ovarian cancers are characterized by an increased risk of developing breast cancer in both men and women, as well as ovarian, fallopian tube, and primary peritoneal cancers [1]. Germline mutations in *BRCA1/2* are responsible for 10–15% of ovarian cancer cases [2]. Additionally, the cumulative risk of ovarian cancer by the age of 80 years is estimated to be 44% (95% confidence interval [CI]: 36–53%) for *BRCA1* carriers and 17% (95% CI: 11–25%) for *BRCA2* carriers [3].

Mutations in *BRCA1/2* are critical predictors of sensitivity to platinum-based chemotherapy [4]. The absence of functional BRCA proteins impairs the ability of cells to repair DNA damage caused by intra-strand crosslinks from DNA cross-linking agents such as platinum drugs. This impaired ability of *BRCA*-deficient tumor cells confers increased sensitivity to chemotherapy. Moreover, PARP inhibition induces synthetic lethality in *BRCA*-mutated cancer cells by preventing the repair of single-strand breaks, leading to double-strand break formation and replication fork collapse, processes that rely on homologous recombination [5]. The use of PARP inhibitors (PARPi) has significantly improved survival benefits in patients with homologous recombination deficiency (HRD), including those with *BRCA1* or *BRCA2* mutations [6,7,8].

*BRCA1* and *BRCA2* are large genes, with exon 11 comprising a significant portion of their structure [9,10]. In particular, women with mutations in *BRCA1* (located on chromosome 17q21) or *BRCA2* (located on chromosome 13q12.3) have an increased risk of developing breast and ovarian cancer. Exon 11 of *BRCA1*, one of the largest exons in the human genome, spans 3426 bases and accounts for approximately 60% of the coding region of the BRCA1 protein. This exon plays a critical role in DNA repair, cell growth, and cell cycle regulation, emphasizing its importance in maintaining genomic stability [11,12,13]. Similarly, *BRCA2* encodes a 3418 amino-acid protein that is essential for genomic stability through its role in repairing DNA double-strand breaks. Exon 11 of *BRCA2*, a large exon similar to *BRCA1*, has also been highlighted for its significant contribution to cancer risk [14].

Mutations in exon 11 of both *BRCA1* and *BRCA2* have drawn considerable attention because of their potential impact on ovarian cancer outcomes, particularly disease onset and progression. Studies have shown that mutations occurring beyond exon 11 of *BRCA1* are associated with a 20% lower risk of ovarian cancer than mutations within exons 1–11 [9]. Mutations within exon 11 of *BRCA2* have been linked to a higher risk of ovarian cancer than of breast cancer [10]. These associations underscore the distinct role of exon 11 in modulating cancer risk. A possible explanation for these findings is that mutations in exon 11 may evade nonsense-mediated mRNA decay (NMD) because of the size of the exon, unlike smaller exons that readily activate NMD [15]. NMD is a cellular surveillance mechanism that typically eliminates mRNAs with premature termination codons (PTCs) and prevents the production of harmful shortened proteins. Ware et al. demonstrated that NMD can effectively recognize PTCs up to 4.5 kb upstream of the nearest exon-exon junction, suggesting that large exons such as exon 11 are not susceptible to NMD activity [14]. Thus, the improved oncological outcomes in patients with mutations in *BRCA* exon 11 are attributed to the large size of exon 11, which enhances platinum sensitivity by evading NMD activity and increasing sensitivity to PARPis. Although emerging evidence suggests that the location of mutations may influence clinical outcomes [16,17], no discernible pattern has been observed indicating which location of the *BRCA1/2* mutation influences clinical outcomes in ovarian cancer patients with *BRCA1* and *BRCA2* mutations.

Therefore, the present study aimed to address this gap by conducting a systematic review and meta-analysis to examine the prognostic implications of *BRCA1/2* mutation locations in epithelial ovarian cancer.

## 2. Materials and Methods

### 2.1. Subsection Literature Search and Selection Criteria

The systematic review and meta-analysis were registered on PROSPERO (PROSPERO 2025 CRD42025632713) and were aligned with the Preferred Reporting Items for Systematic Reviews and Meta-Analyses (PRISMA) guidelines [18]. A comprehensive literature search was performed using three major electronic databases: PubMed, Embase, and the Cochrane Library. The search was limited to studies published in English and included all the relevant literature published up to 13 August 2024. The following combinations of keywords were used to identify relevant articles, and the search formula is shown in Appendix A. A total of 4293 studies were identified after excluding duplicate studies. Studies were screened based on the following exclusion criteria: (1) non-English language, (2) no human data, (3) no relevant data on the location of the *BRCA* mutation, (4) no survival data, (5) unavailability of full-text or abstract, and (6) other cancers. Additionally, we excluded studies with irrelevant data and those lacking sufficient data on survival outcomes based on the location of *BRCA* mutations (Figure 1).

### 2.2. Study Selection, Data Extraction

Two independent investigators, JHK and HJY, conducted a comprehensive review of all identified studies. Any discrepancies encountered were resolved through consensus. For the meta-analysis, we selected progression-free survival (PFS) and overall survival (OS) as endpoints, based on the exon 11 mutation status of *BRCA1/2*, irrespective of the presence of mutations in other exons, as such information was not consistently available across the studies. The following data were obtained from each study: authors, year of publication, study design, patient count, median follow-up duration, hazard ratios (HRs) for PFS and OS, with 95% CIs, median age, histology, and stage, according to the International Federation of Gynecology and Obstetrics (FIGO). The quality of each study was rated using the Newcastle–Ottawa scale (NOS).

### 2.3. Statistical Analysis

HR was used as a measure of prognostic value. HR > 1 indicated poor survival in the group with exon 11 mutations in *BRCA 1* and *2*. HRs and 95% CIs were extracted from the articles. The hazard and standard error (SE) were estimated using median survival time and survival probability, assuming an exponential distribution in the case of missing HR and 95% CI values. The exponential distribution is coded as follows:Hazard rate (h): h = log(2)/M or −log(P)/T, where M is the median survival time, P is the survival probability, and T is time (years);Hazard ratio (HR) = h1/h;Standard error (SE) for log(HR): (log(upper CI) − log(lower CI))/(2 × 1.96) or sqrt(1/expected events1 + 1/expected events2), where expected events = 1/hazards.

The results, excluding the estimates, are included in Appendix A. Statistical analyses were performed using the R statistical language (version 4.4.1; R Core Team, 2024) and additional packages (meta). All tests were two-sided, with a significance level of *p* < 0.05. Heterogeneity was expressed using the I^2^ index [19], which describes the percentage of total variation across studies that is due to heterogeneity rather than chance (25%, low heterogeneity; 50%, medium; and 75%, high).

Publication bias was assessed visually through funnel plots for progression-free survival (PFS) and overall survival (OS). The symmetry of the funnel plots was evaluated to ascertain the potential for small-study effects or publication bias.

This study is a systematic review and meta-analysis using previously published data. Therefore, formal Ethical or Institutional Board Approval or Exemption was not required. In accordance with the journal’s guidelines, we will provide our data for independent analysis by a selected team by the Editorial Team for the purposes of additional data analysis or the reproducibility of this study in other centers if such is requested.

## 3. Results

### 3.1. Study Characteristics

Table 1 summarizes the selected studies. We screened 6712 articles and identified seven studies deemed suitable for analysis, which were included in the meta-analysis. All selected studies had a retrospective design. The median patient age ranged from 51 to 59 years. Most patients presented with advanced-stage disease (FIGO stages III–IV), with the percentage of serous histology ranging from 45.6 to 94.2%. Most studies reported the use of platinum-based chemotherapy as the standard treatment regimen, with some studies including patients treated with PARPis. The methodological quality of the included studies, as assessed using the NOS, varied but was generally high. Most studies received high scores, indicating strong methodological rigor and a low risk of bias. However, few studies reported insufficient data on specific aspects, such as the adequacy of follow-up.

### 3.2. BRCA 1/2 Exon 11 Mutations vs. Non-BRCA 1/2 Exon 11 Mutations: PFS and OS

Figure 2 shows that the investigation of *BRCA1* and *BRCA2* exon 11 mutations compared to non-exon 11 mutations revealed no significant differences in oncological outcomes. The HR for PFS was 0.717 (95% CI: 0.467–1.102), suggesting no significant benefit among the mutation types. Similarly, OS analysis revealed an HR of 0.494 (95% CI: 0.192–1.270) with no statistically significant difference. These findings suggest that *BRCA 1/2* exon 11 mutations do not provide significant benefits in terms of PFS and OS relative to non-*BRCA 1/2* exon 11 mutations. However, subgroup analysis showed significant differences between *BRCA1* and *BRCA2* exon 11 mutations.

### 3.3. BRCA 1 Exon 11 Mutations vs. Non-BRCA 1 Exon 11 Mutations: PFS and OS

Subgroup analysis of *BRCA1* exon 11 and non-exon 11 mutations revealed no statistically significant differences in the PFS or OS (Figure 3). The pooled hazard ratio for PFS was 0.823 (95% CI: 0.321–2.109), indicating no substantial benefit in PFS for *BRCA1* exon 11 mutations. The pooled HR for OS was 0.953 (95% CI: 0.557–1.632), demonstrating no significant differences in the survival outcomes. Although statistical significance was absent, considerable heterogeneity was observed in the PFS analysis (I^2^ = 85.9%, *p* < 0.001), indicating variation among the included studies. Conversely, the OS analysis for *BRCA1* exon 11 mutations showed minimal heterogeneity (I^2^ = 7.6%, *p* = 0.298), indicating more uniform results. These results suggest that *BRCA1* exon 11 mutations may not play a critical role in the oncologic outcomes of ovarian cancer compared to non-*BRCA1* exon 11 mutations, but the heterogeneity in PFS warrants cautious interpretation.

### 3.4. BRCA 2 Exon 11 Mutations vs. Non-BRCA 2 Exon 11 Mutations: PFS and OS

In contrast to *BRCA1*, *BRCA2* exon 11 mutations demonstrated significant differences in PFS compared to non-*BRCA2* exon 11 mutations (Figure 3). The pooled HR was 0.586 (95% CI: 0.346–0.994, *p* < 0.05), indicating a significant PFS benefit for *BRCA2* exon 11 mutations. In terms of OS, the pooled HR was 0.035 (95% CI: 0.000–5.270), indicating no OS benefit for *BRCA2* exon 11 mutations. However, the extremely high heterogeneity observed in OS (I^2^ = 99.5%, *p* < 0.001) limited the reliability of the results. Thus, these results highlight the significant clinical relevance of *BRCA2* exon 11 mutations in improving oncological outcomes. However, the substantial heterogeneity among studies shows variability in OS outcomes, making definitive conclusions challenging. This underscores the need for further studies to validate these findings.

### 3.5. Publication Bias Analysis

To evaluate the probability of publication bias, we developed funnel plots for the analyses of progression-free survival (PFS) and overall survival (OS) (Appendix A). The funnel plots exhibited considerable symmetry, indicating a lack of substantial evidence for publication bias in the studies analyzed.

## 4. Discussion

To the best of our knowledge, this meta-analysis is the first systematic evaluation of the impact of mutations in specific *BRCA* exons, particularly exon 11, on the PFS and OS of patients with ovarian cancer. Our results demonstrate the clinical significance of *BRCA* exon 11 mutations and provide insight into targeted therapeutic strategies.

The results in Figure 2 indicate hazard ratios that suggest enhanced progression-free survival (PFS) and overall survival (OS) in patients with *BRCA1/2* exon 11 mutations compared to those with non-exon 11 mutations. However, it is important to note that these differences did not reach statistical significance. A meta-analysis by Tobalina et al. found that exon 11 mutations in *BRCA1* and *2* did not significantly impact the development of resistance to platinum chemotherapy or PARPis [18]. Labidi-Galy et al. performed a subgroup analysis of PAOLA-1 data to explore the therapeutic effects of olaparib and bevacizumab maintenance according to the location of *BRCA 1/2* mutations in patients with high-grade ovarian cancer [19]. They found that patients with *BRCA1/2* exon 11 mutations had greater benefit from PARPi treatment compared to patients with non-exon 11 mutations [exon 11 mutation group HR 0.2 (95% CI: 0.11–0.36), non-exon 11 mutations group HR 0.41 (95% CI: 0.22–0.75)]. Although no statistical significance, our finding aligns with these observations and suggests that the location of *BRCA1/2* mutations may influence the treatment response and prognosis in complex ways.

The observed PFS advantage in patients with *BRCA2* exon 11 mutations (HR, 0.586; 95% CI, 0.346–0.994) could be attributed to the RAD51-binding domain in exon 11. RAD51 is essential for homologous recombination, a critical DNA repair mechanism [20]. Mutations in *BRCA2* exon 11 can disrupt the interaction between BRCA2 and RAD51, leading to impaired homologous recombination and increased reliance on alternative error-prone repair pathways. This deficiency enhances the cytotoxic effects of platinum-based chemotherapy, which induces DNA crosslinking and double-stranded breaks that require functional homologous recombination for repair. Consequently, tumors harboring *BRCA2* exon 11 mutations exhibit heightened sensitivity to such treatments, resulting in better clinical outcomes. These findings align with evidence that *BRCA2*-mutated cases exhibit significantly higher chemotherapy sensitivity rates and longer platinum-free durations than *BRCA1*-mutated and wild-type *BRCA* cases [21]. A previous study reported that *BRCA2*-mutated tumors demonstrate a “mutator phenotype”, characterized by a higher mutation burden, which may contribute to their enhanced response to platinum-based treatment. In contrast, *BRCA1*-mutated cases do not show such associations, suggesting that the differing prognostic effects of *BRCA1* and *BRCA2* deficiencies likely stem from distinct mechanisms of dysfunction and chemotherapy responses [22].

Unlike *BRCA2* exon 11 mutations, *BRCA1* exon 11 does not appear to affect PFS when mutated significantly, despite being the largest exon and contributing to DNA repair. This result may be due to the multifunctional nature of *BRCA1*, which contains other critical domains such as the *RING* and *BRCT* domains that facilitate interactions with various proteins involved in DNA damage response and repair [23,24]. These domains may compensate for the loss of function caused by exon 11 mutations, thereby mitigating their impact on treatment outcomes. Notably, the limited clinical benefit of *BRCA1* exon 11 mutations may partly be explained by alternative splice isoforms, such as *BRCA1*-Δ11q, which bypass germline mutations and promote resistance to PARPis and cisplatin [25]. Additionally, *BRCA1* secondary splice-site mutations have been identified as a mechanism of PARPi resistance through exon skipping and overexpression of *BRCA1* hypomorphs, further highlighting the complexity of *BRCA1*-associated resistance pathways [26].

In this meta-analysis, *BRCA 1* exon 11 mutations were found to have no benefit on PFS and OS, which contradicts the results by Ha et al. [17]. They compared 79 patients with PV within the OCCR among 162 *BRCA1* PV patients with 83 patients with *BRCA1* PV within the non-OCCR and found that patients with PV within the ovarian cancer cluster region (OCCR) had worse PFS (*p* = 0.038). This trend was more pronounced in the platinum-sensitive group. This discrepancy in the results may be due to two reasons. First, the OCCR proposed by Rebbeck et al. is similar to, but not identical to, exon 11 [27]. Ha et al. identified an OCCR from c.1380 and c.4062. In addition, because the concept of OCCR itself is estimated using statistical methods rather than being divided by the structure or function of the protein, its correlation with the functional region in exon 11 is unknown. Second, they included patients who were diagnosed with primary peritoneal, ovarian, and fallopian tube cancers between 1 January 2006, and 31 August 2019. Additionally, recent developments have led to major changes in the treatment of ovarian cancer, including the introduction of PARPis and bevacizumab, and the expanded use of HIPEC; however, the effect of these changes in treatment on patient survival remains unknown. Third, it is essential to acknowledge that clinical outcomes in patients with *BRCA*-mutated cancer may also be affected by treatment-related factors. For instance, a recent study involving *BRCA* mutation carriers with breast cancer revealed that surgical decisions, such as opting for breast-conserving therapy versus mastectomy and the inclusion of bilateral salpingo-oophorectomy (BSO), significantly influenced distant disease-free and overall survival [28]. This indicates that, in addition to the mutation location, the treatment strategy may play a pivotal role in determining oncologic outcomes. Therefore, additional research is necessary to determine whether clinical outcomes change according to the location of the pathogenic variant in *BRCA1* or *BRCA2*.

While this study yields positive outcomes, it is not without limitations. Firstly, the retrospective nature of the included studies may have introduced selection bias. Secondly, the significant heterogeneity observed in certain analyses, particularly overall survival (OS) for *BRCA2* exon 11, constrains the generalizability of the findings. The high degree of heterogeneity in the OS analysis for *BRCA2* exon 11 mutations could not be adequately explained, and the limited number of studies precluded any reliable meta-regression, thereby significantly limiting the interpretability of the pooled estimate. Thirdly, confounding variables—such as treatment era, PARPi usage, and patient characteristics—could not be fully assessed due to the reliance on published data. Furthermore, we were unable to classify mutations by functional type (e.g., missense, nonsense, or frameshift) due to the inconsistent reporting of this information across the included studies. Considering that different mutation types may exhibit varying pathogenicity and biological effects, this limitation may have contributed to unmeasured heterogeneity. We recommend that future studies report outcome data stratified not only by mutation type but also by treatment modality, particularly the use of PARP inhibitors, to better elucidate the prognostic implications of *BRCA* exon 11 mutations. Notably, the limited number of eligible studies highlights a significant gap in the current literature and underscores the necessity for further research specifically addressing exon-level mutation impacts on survival in *BRCA*-mutated ovarian cancer.

## 5. Conclusions

Our results indicate that the location of *BRCA* mutations, particularly within *BRCA2* exon 11, is associated with differences in the survival outcomes of patients with ovarian cancer. This highlights the clinical relevance of *BRCA2* exon 11 mutations and emphasizes their association with improved PFS. These findings underscore the biological significance of exon 11 by consolidating evidence from multiple studies that suggest potential prognostic implications of mutations within this region, thereby providing a foundation for future research despite the heterogeneity among studies. Consequently, further prospective studies are warranted to confirm these associations and to investigate the potential impact of other BRCA mutation locations on clinical outcomes.

## Figures and Tables

**Figure 1 cancers-17-01661-f001:**
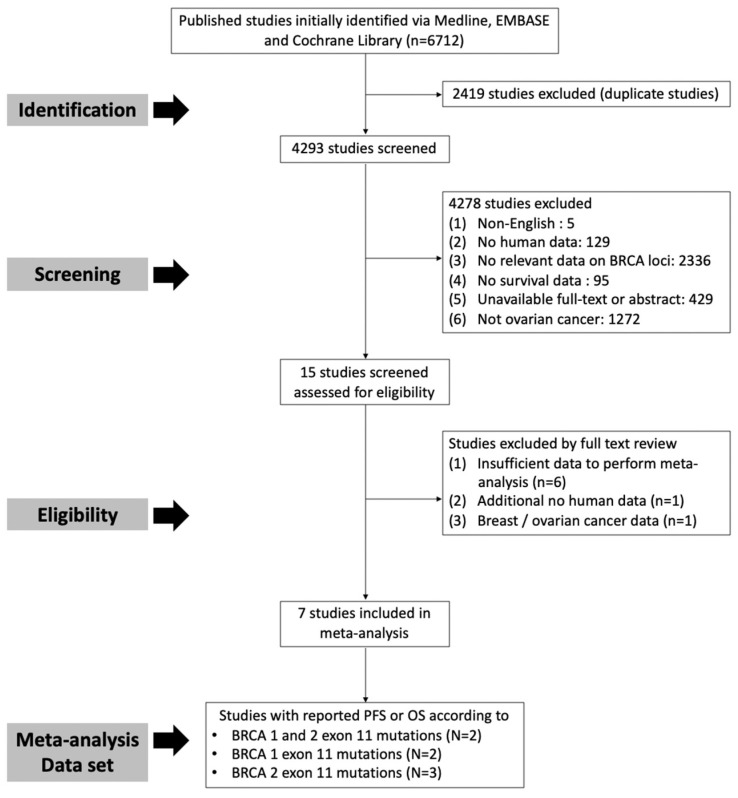
Flow diagram for selection of studies.

**Figure 2 cancers-17-01661-f002:**
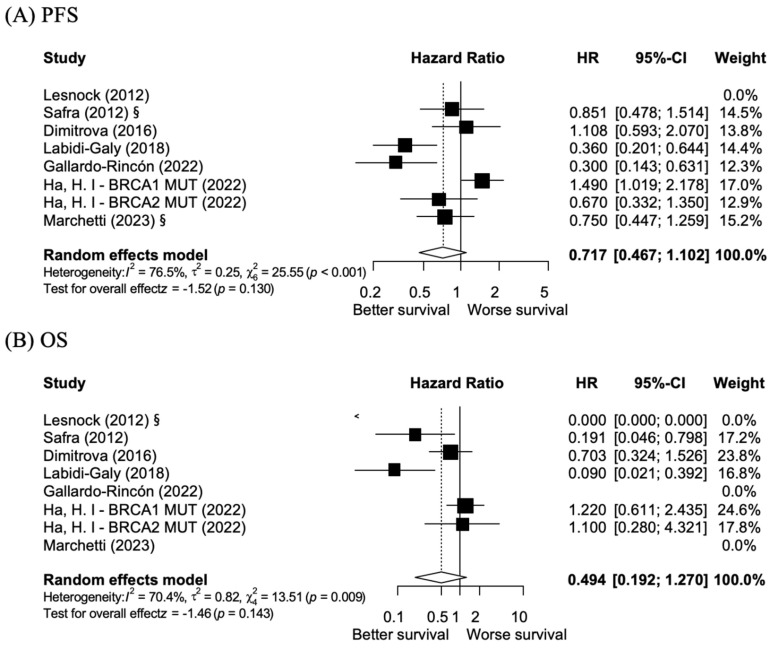
Forest plot of the association between the *BRCA 1/2* exon 11 mutation and non-exon 11 mutation in ovarian cancer patients. (**A**) Forest plot of Progression-Free Survival (PFS) comparing *BRCA1/2* exon 11 mutations to non-exon 11 mutations in ovarian cancer patients. (**B**) Forest plot of overall survival (OS) comparing *BRCA1/2* exon 11 mutations to non-exon 11 mutations in ovarian cancer patients. “§” Estimated values based on exponential distribution. “<” represents that the plotted HR lies beyond the lower bound of the x-axis scale due to an estimated HR of zero and weight of 0.0%. (References: Lesnock [20], Safra [21], Dimitrova [22], Labidi-Galy [23], Gallardo-Rincón [24], Ha et al. [17], Marchetti [25]).

**Figure 3 cancers-17-01661-f003:**
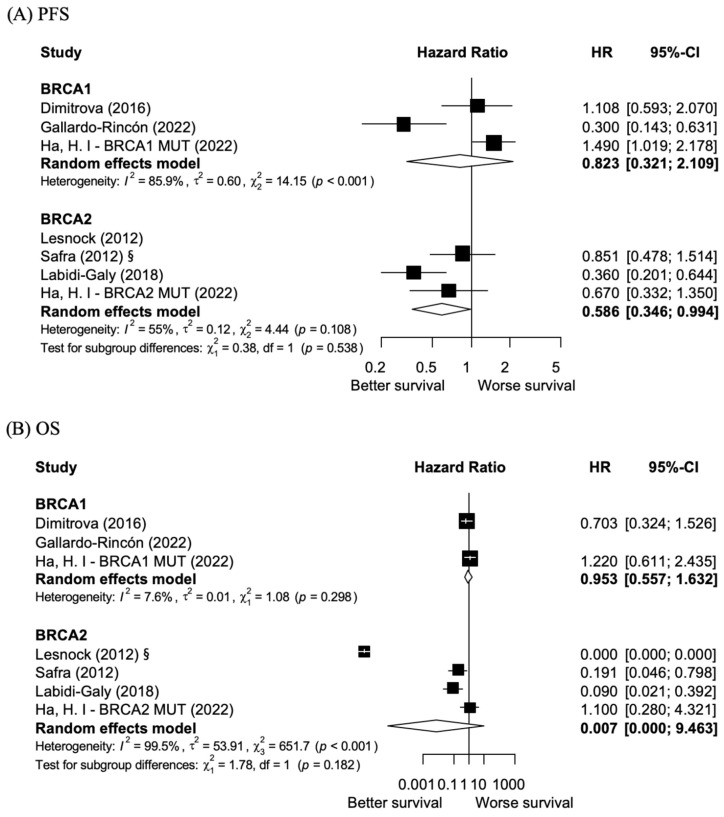
Subgroup Analysis: Forest Plots of Progression-Free Survival (PFS) and overall survival (OS) for *BRCA1* and BRCA2 exon 11 vs. non-exon 11 mutations. (**A**) Forest plot of Progression-Free Survival (PFS) for *BRCA1* and *BRCA2* subgroups. (**B**) Forest plot of overall survival (OS) for *BRCA1* and *BRCA2* subgroups. “§” Estimated values based on exponential distribution. (References: Lesnock [20], Safra [21], Dimitrova [22], Labidi-Galy [23], Gallardo-Rincón [24], Ha et al. [17], Marchetti [25]).

**Table 1 cancers-17-01661-t001:** Characteristics of the included studies.

No.	Author	Year	Total (N)	Mutation (N)	Control (N)	BRCA1/2 Mutation	Median Follow-Up (Months)	PFS HR	CI	*p*-Value	OS HR	CI	*p*-Value	Median Age	FIGO Stage III–IV	Histology (Serous)	Chemotherapy	PARPis	NOS	Remarks
1	Lesnock, J. et al. [20]	2012	315	5	56	BRCA2	35.4												8/9	poster
2	Safra, T. et al. [21]	2012	190	12	100	BRCA2	56	0.851	0.478–1.514	0.583	0.191	0.05–0.798	0.023	55.5	169 (88.9%)	132 (69.5%)	(+)	(−)	8/9	
3	Dimitrova, D. et al. [22]	2016	263	18	235	BRCA1	51	1.108	0.593–2.070	0.748	0.703	0.33–1.53	0.373	56	188 (71.5%)	211 (80.2%)	(+)	(−)	8/9	
4	Labidi-Galy, S. I. et al. [23]	2018	353	42	36	BRCA2	48	0.360	0.201–0.644	0.001	0.090	0.02–0.39	0.001	59	222 (23.5%)	180 (67.7%)	(+)	(−)	8/9	
5	Gallardo-Rincón, D. et al. [24]	2022	35	9	26	BRCA1	12.87	0.300	0.143–0.631	0.002				51	31 (88.6%)	33 (94.2%)	(+)	(+)	7/9	
6	* Ha, H. I. et al.—BRCA1 [17]	2022	238	79	83	BRCA1	25.5	1.490	1.019–2.178	0.040	1.220	0.61–2.44	0.573	54	71 (91.1%)	36 (45.6%)	(+)	(−)	9/9	
7	* Ha, H. I. et al.—BRCA2 [17]	2022	238	27	49	BRCA2	25.5	0.670	0.332–1.35	0.263	1.100	0.28–4.32	0.891	58	24 (88.9%)	13 (48.1%)	(+)	(−)	8/9	
8	Marchetti, C. et al. [25]	2023	141			BRCA1/2	20	0.750	0.447–1.259	0.277								(+)	9/9	poster

* Note: The study by Ha et al. (2022) [17] reported separate data for BRCA1 and BRCA2 exon 11 mutations; both outcomes are listed individually for clarity. PFS: Progression-Free Survival; HR: hazard ratio; CI: confidence interval; OS: overall survival; NOS: Newcastle–Ottawa scale.

## Data Availability

No new data were created or analyzed in this study. Data sharing is not applicable to this article.

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
