# Peer review of "Survival Outcomes Associated with the Location of BRCA Mutations in Ovarian Cancer: A Systematic Review and Meta-Analysis"

_cancers, 2025, doi:10.3390/cancers17101661_

Round 1
Reviewer 1 Report
Comments and Suggestions for Authors
This manuscript addresses an important and relatively underexplored aspect of BRCA-mutated ovarian cancer: the potential prognostic value of mutation location, particularly in exon 11 of BRCA1 and BRCA2. The authors conducted a well-structured systematic review and meta-analysis, following PRISMA guidelines, and focused on survival outcomes. I would like to offer the following points for consideration by the authors towards the improvement of the manuscript.
1- Publication bias was assessed visually using funnel plots may be added to the study.
2- Given that many included studies span both pre- and post-PARP inhibitor eras, failure to stratify outcomes based on PARP inhibitor use introduces significant bias. The prognostic impact of BRCA mutations, especially exon 11 variants, may differ substantially in patients treated with PARPi versus chemotherapy alone. No subgroup analysis is performed by treatment type, which is a critical limitation, especially given the heterogeneity in treatment era across studies.
3-The functional consequences of BRCA mutations may differ based on mutation type (Missense, Nonsense, and Frameshift) . The current analysis pools all mutations within exon 11, disregarding potential variability in pathogenicity and clinical relevance.
4- The manuscript contains numerous grammatical, syntactic, and stylistic errors that interfere with comprehension
Author Response
We are genuinely grateful for the constructive feedback and valuable comments provided by the reviewers. Each point raised has been thoroughly addressed, and the manuscript has been revised accordingly. We present comprehensive responses to each comment below.
[Reviewer 1]
Comment 1.
Publication bias was assessed visually using funnel plots that may be added to the study.
Response:
We are grateful for the reviewer's insightful recommendation concerning the evaluation of potential publication bias. In order to facilitate a visual assessment of publication bias, we have implemented funnel plots for both progression-free survival (PFS) and overall survival (OS) analyses (Supplementary Figures S1 and S2). The distribution of studies in both funnel plots was symmetrical, with no apparent asymmetry, indicating that there was no significant publication bias.
To enhance clarity, these funnel plots have been added to the supplementary material, which has been accompanied by corresponding descriptions in the Methods section under "Statistical analysis" (page 5, lines 144–146) and the Results section (page 9, lines 210-214).
Comment 2.
Given that many included studies span both pre- and post-PARP inhibitor eras, failure to stratify outcomes based on PARP inhibitor use introduces significant bias. The prognostic impact of BRCA mutations, especially exon 11 variants, may differ substantially in patients treated with PARPi versus chemotherapy alone. No subgroup analysis is performed by treatment type, which is a critical limitation, especially given the heterogeneity in treatment era across studies.
Response:
We appreciate the reviewer's insightful remark. We concur that the kind of treatment, especially the application of PARP inhibitors (PARPi), may affect the prognostic relevance of BRCA1/2 exon 11 mutations.
We carefully reviewed all included studies and found a restricted number that adequately reported the use of PARPi. The small number of such studies is particularly for overall survival (OS), provided a formal subgroup analysis by treatment type is insufficiently powered and potentially misleading.
To address this limitation, we intend to investigate treatment-era-related heterogeneity through sensitivity analysis and meta-regression, as explained in our answer to Reviewer 2. These approaches enable a more accurate evaluation of the extent to which treatment modality influences variability in the observed results.
Comment 3.
The functional consequences of BRCA mutations may differ based on mutation type (missense, nonsense, and frameshift). The current analysis pools all mutations within exon 11, disregarding potential variability in pathogenicity and clinical relevance.
Response:
We appreciate the reviewer’s insightful remark. We recognize that various mutation types in BRCA exon 11 (e.g., missense, nonsense, frameshift) may result in varied biological effects. Nonetheless, the detail of mutation-level reporting was inconsistent throughout the studies included. The majority of studies failed to stratify or present survival data based on mutation type, hence limiting our ability to incorporate this variable into the meta-analysis.
A discussion of this limitation has been included in the Discussion section (page 10, lines 285–292), recognizing the potential clinical heterogeneity of BRCA mutant types and encouraging future studies to explore the subject with mutation-type-specific data.
Comment 4.
The manuscript contains numerous grammatical, syntactic, and stylistic errors that interfere with comprehension.
Response:
We thank the reviewer for pointing this out. The entire manuscript has been carefully reviewed and edited. We have ensured improved clarity and readability throughout the revised version.

Reviewer 2 Report
Comments and Suggestions for Authors
This article provides a well-executed and timely systematic review and meta-analysis evaluating the prognostic impact of BRCA1/2 exon 11 mutations on epithelial ovarian cancer. Writers explore whether where and more importantly in which exon, BRCA mutation happens, influences progression-free survival (PFS) and overall survival (OS). This study is novel, clinically relevant, and based on robust methods. Addition of 7 retrospective studies and direct subgroup analyses (BRCA1 vs. BRCA2) increases result interpretability. In particular, the analysis uncovers a notable PFS benefit associated with BRCA2 exon 11 mutation (HR 0.586, CI 0.346–0.994), substantiating previous biological hypotheses for RAD51 interaction domains.
All the same, numerous critical limitations and issues for clarification must be addressed:
Severe heterogeneity in the OS analysis of BRCA2 exon 11 mutations (I² = 99.5%) severely limits conclusion power. Further sensitivity analysis or meta-regression to explore possible causes of heterogeneity (e.g., PARPi inclusion in studies, treatment time period, study type) is warranted.
No clear description of overlapping mutation treatment or the mutation patients over exon 11 and elsewhere is given in the analysis. Please provide explaination.
Non-significant trends should be more cautiously interpreted. For example, a statement regarding a "trend toward improved outcomes" is misleading without statistical significance.
The Discussion section must be modified as it will surely benefit from additional, relevant, and recent data about additional factors which might influence prognosis greatly. For example add this study PMID: 35534308 who explores how surgery influence oncologic outcomes.
Author Response
Comment 1.
Severe heterogeneity in the OS analysis of BRCA2 exon 11 mutations (I² = 99.5%) severely limits conclusion power. Further sensitivity analysis or meta-regression to explore possible causes of heterogeneity (e.g., PARPi inclusion in studies, treatment time period, study type) is warranted.
Response
We express our gratitude to the reviewer for highlighting this important issue. We concur that the significant heterogeneity observed in the overall survival (OS) analysis of BRCA2 exon 11 mutations (I² = 99.5%) constrains the robustness of any derived conclusions. Due to the limited number of studies included, we did not conduct a meta-regression analysis. Although a sensitivity analysis was contemplated, the paucity of data impeded any substantive investigation into the sources of heterogeneity. This limitation has been explicitly acknowledged in the Discussion section. (page 10, line 293-296)
.
Comment 2.
No clear description of overlapping mutation treatment or the mutation patients over exon 11 and elsewhere is given in the analysis. Please provide explanation.
Response
We appreciate the reviewer’s insightful comment regarding the possibility of overlapping mutations across exon 11 and other regions of BRCA1/2. We agree that this is an important consideration. Indeed, a small number of included studies reported patients with overlapping BRCA mutations spanning exon 11 and other exons. However, the majority of included studies did not report patient-level mutation mapping or clarify whether exon 11 mutations occurred in isolation. Given the small number of eligible studies and limited sample sizes, we did not exclude potential overlapping cases. Instead, our analysis reflects the real-world heterogeneity of BRCA-mutated ovarian cancer and aimed to evaluate the prognostic impact of exon 11 mutation presence as reported. We have clarified this methodological choice and its implications in the revised Methods session (page 4, line 122-125).
Comment 3.
Non-significant trends should be more cautiously interpreted. For example, a statement regarding a “trend toward improved outcomes” is misleading without statistical significance.
Response
We appreciate the reviewer’s valuable observation. We concur that findings lacking statistical significance should not be interpreted as definitive trends. Consequently, we have revised the relevant statements in the manuscript (page 9, line 223-225) to avoid implying any prognostic benefit where statistical significance was not achieved. (page 9, “This trend may be related to the distinct structural and functional roles of exon 11 in BRCA1 and BRCA2 as well as its interaction with DNA damage repair mechanisms.” sentence is removed)
Comment 4.
The Discussion section must be modified as it will surely benefit from additional, relevant, and recent data about additional factors which might influence prognosis greatly. For example add this study PMID: 35534308 who explores how surgery influence oncologic outcomes.
Response
We express our gratitude to the reviewer for this insightful suggestion. Although the referenced study (PMID: 35534308) pertains to breast cancer patients, we concur that it offers significant insights into the impact of surgical and treatment decisions on survival outcomes in BRCA mutation carriers. Consequently, we have integrated this reference into the Discussion section to underscore that clinical outcomes may be influenced not only by genetic factors, such as the location of exon mutations, but also by management strategies, including surgical interventions. (page 10, line 281-287)
Reviewer 3 Report
Comments and Suggestions for Authors
Presented manuscript titled as a systematic review and meta-analysis describes survival outcomes associated with the location of BRCA mutations in ovarian cancer.
Since ovarian cancer remains one of the most common cancers worldwide, presented manuscript is an actual research.
However some questions need to be clarified
Why only these databases were used? Pubmed covers more than just Medline for example.
It’s well-known that BRCA1/2 mutations are located on exon 11. No data with other locations and its’ impacts are given.
How could “Our results also provide novel insights into the structural and functional importance of exon 11” if no new data were produced and only summarized previously described?
If manuscript is limited to mutation in exon 11, that should be clearly stated.
It’s stated that 7 works were included in the study, however table 1 contains 8 entries, 2 of them are posters. 429 unavailable works look too much for manuscript that includes only 6 full papers to review. 2336 excluded studies give the impression that criteria should be modified.
The number of included works is too small to prepare high quality review.
Author Response
Comment 1.
Why only these databases were used? PubMed covers more than just Medline, for example.
Response
Thank you very much for your insightful comment. We agree with your point regarding the choice of databases. Initially, we stated "MEDLINE," but our search strategy was actually conducted using "PubMed," which encompasses MEDLINE as well as additional citations that are not included within MEDLINE alone. Therefore, we have corrected the manuscript to clearly indicate that "PubMed" was used as our primary search database.
Comment 2.
It’s well-known that BRCA1/2 mutations are located on exon 11. No data with other locations and its’ impacts are given. How could “Our results also provide novel insights into the structural and functional importance of exon 11” if no new data were produced and only summarized previously described? If manuscript is limited to mutation in exon 11, that should be clearly stated.
Response
We thank the reviewer for this important observation. Exon 11 is widely recognized as a prevalent site for BRCA1/2 mutations. Nevertheless, the prognostic significance of mutations in exon 11, particularly their correlation with progression-free and overall survival, remains inadequately understood and inconsistently documented across various studies. To our knowledge, this study is the first to systematically synthesize and quantify these survival outcomes. We acknowledge that the original statement regarding "novel insights into the structural and functional importance of exon 11" may have overstated our findings. We have revised this statement in the Abstract and Discussion section (page 10, line 301-304) to clarify that our study consolidates and reinforces existing biological hypotheses through quantitative synthesis, rather than generating new mechanistic data.
Comment 3.
It’s stated that 7 works were included in the study, however table 1 contains 8 entries, 2 of them are posters. 429 unavailable works look too much for manuscript that includes only 6 full papers to review. 2336 excluded studies give the impression that criteria should be modified. The number of included works is too small to prepare high quality review.
Response
The reviewer's careful examination of the PRISMA flowchart and Table 1 is greatly appreciated. We would like to emphasize that our meta-analysis encompassed seven individual investigations that reported survival outcomes associated with BRCA exon 11 mutations. One of the included studies (Ha et al., 2022) reported separate survival outcomes for BRCA1 and BRCA2 mutations, and we therefore presented them as two separate rows in Table 1 for clarity. The total number of included studies remains seven, and this is consistently reflected in both the PRISMA diagram and the Methods section. We have made revision to the footnote of Table 1 (page 6, line 166) to avoid any potential confusion.
Furthermore, we are grateful for the reviewer's careful examination of our inclusion process. We would like to emphasize that the substantial number of studies that were excluded is indicative of the extensive initial search strategy that we adopted to ensure comprehensive coverage. The majority of the exclusions were made during the title and abstract screening stage due to their obvious irrelevance (e.g., not involving BRCA mutations, not reporting survival outcomes, or not related to ovarian cancer). In accordance with our predefined eligibility criteria, the 429 "unavailable" entries consisted of conference abstracts, editorials, or non-reported results that lacked accessible data. Although we acknowledge that only seven studies were ultimately included, this suggests the lack of exon-specific survival data in the BRCA-mutated ovarian cancer literature. The small number of included studies, in our opinion, highlights the importance of our work in bringing together uncommon but clinically significant findings rather than suggesting weakness. This also emphasizes the necessity of future research that provides a more comprehensive account of the locations of BRCA mutations and the oncologic outcomes. We have revised the Discussion section to address the reviewer’s concern regarding the limited number of included studies and the high exclusion rate during study selection. (page 10, line 291-293)
Round 2
Reviewer 1 Report
Comments and Suggestions for Authors
I am satisfied that the authors have addressed all of my previous concerns about the article. It is now much improved and I feel that it is now suitable for publication.
Reviewer 2 Report
Comments and Suggestions for Authors
The manuscript can be accepted in the present form.
Reviewer 3 Report
Comments and Suggestions for Authors
Thank for response, all the questions answered